

# Uncertainty budget in snow thickness and snow water equivalent estimation using GPR and TDR techniques

Federico Di Paolo[1], Barbara Cosciotti[1], Sebastian E. Lauro[1], Elisabetta Mattei[1], Mattia Callegari[2], Luca Carturan[3], Roberto Seppi[4], Francesco Zucca[4], Elena Pettinelli[1]

[1]Department of Mathematics and Physics, Roma Tre University, Rome, 00146, Italy
[2]Institute for Applied Remote Sensing, EURAC, Bolzano, 39100, Italy
[3]Department of Land, Environment, Agriculture and Forestry (TeSAF), University of Padova, Legnaro (PD), 35020, Italy
[4]Department of Earth and Environmental Sciences, University of Pavia, Pavia, 27100, Italy

*Correspondence to*: Elena Pettinelli (pettinelli@fis.uniroma3.it)



**Abstract.** Snow water equivalent is a fundamental parameter for hydrological and climate change studies but its measurement is usually time consuming and destructive. Electromagnetic methods could be a valid alternative to conventional techniques, being fast and non-invasive. In this work we analyze the reliability of a combined GPR/TDR method to estimate snow thickness and snow water equivalent. To estimate GPR accuracy we perform a calibration test

where measured and predicted radar data are compared in terms of two-way travel time. Furthermore we implement a complete analysis of the uncertainty budget in order to evaluate the "weight" of each uncertainty on the snow parameters computation chain. We found that GPR, supported by TDR data, is quite reliable as it measures snow thickness and snow water equivalent with an accuracy comparable to that of a traditional method but, in general, with a slightly larger uncertainty.

## 1 Introduction

Snow water equivalent (SWE) is a quantity computed as the product of snow depth and snow density that expresses the thickness of water that would theoretically result if the entire snowpack was instantaneously melted. In a conventional approach, snow thickness is measured using and hand probe (snow rod) and density is usually measured along the wall of a snow pit sampling and weighing the snow (Kinar and Pomeroy, 2015 and references therein).

Several alternatives to such methods have been proposed and tested by various authors, as comprehensively described by a recent literature review article (Kinar and Pomeroy, 2015). Actually, as two different physical parameters are needed (snow thickness and density) it is difficult to find a single instrument capable to properly measure both quantities, thus a strategy combining two different techniques is usually required. The applied techniques should be both accurate and precise to minimize any possible bias affecting the overall evaluation of the snow water equivalent and to provide values that are

affected by the minimum possible uncertainty. Furthermore such techniques should be suitable to cover large areas in a reasonable time and, possibly, should be non-destructive.

One of the most promising geophysical tool to quickly and extensively measure snow thickness is Ground Penetrating Radar (GPR) (e.g., Godio, 2009; Forte et al., 2015), due to the high transparency of snow to radio waves (Annan et al., 1994). GPR provides an electromagnetic image of the subsurface as the result of the interaction between the emitted pulses and the

electromagnetic properties of the materials in the subsurface (Jol, 2008). The transmitting antenna (Tx) sends short pulses into the snow and the receiving antenna (Rx) gathers such pulses after they have been propagated downward and reflected back by any dielectric interface (e.g., snow/ice or snow/rocks) present at depth (Annan et al., 1994). GPR, as any other radar, measures the time it takes for the pulses to make a round trip Tx-reflecting interface-Rx, named two-way travel time. To convert this parameter into snow thickness the electromagnetic velocity of the radar pulses should be known. In principle,

GPR is capable to measure the pulse velocity if used in specific configurations or if appropriate targets are present in the snow (Eisen et al., 2003; Bradford and Harper, 2005; Bradford and Harper, 2006; Bradford et al., 2009; Godio, 2009;





Gustafsson et al., 2012; Forte et al., 2014; Holbrook et al., 2016). In practice, however, there are many cases where these measurements are unfeasible or unreliable like, for example, when a fixed offset configuration is used, thus the evaluation of the wave velocity requires a support from an independent method. Time Domain Reflectometry (TDR) probably represents the best choice as it is conceptually similar to GPR and can be easily applied to snow studies (Stein and Kane, 1983;

Lundberg, 1997; Stein et al., 1997; Schneebeli et al., 1998; Stacheder et al., 2005).

Numerous papers have addressed the issue of snow thickness estimation, density evaluation and snow water equivalent calculation using GPR alone, together with some other classical methods (Annan et al., 1994; Sand and Bruland, 1998; Lundberg et al., 2010; Marchand et al., 2001; Bradford and Harper, 2006; Lundberg et al., 2006; Bradford et al., 2009; Godio, 2009; Gustaffson et al., 2012; Sundström et al., 2012; Forte et al., 2013; Godio and Rege, 2016; Holbrook et al.,

2016) or combining GPR and TDR techniques (Harper and Bradford, 2003; Previati et al., 2011; Di Paolo et al., 2015). However, our prospective is quite different as we are interested in understanding how accurate and precise the estimation of the snow water equivalent performed using GPR (supported by another method for wave velocity estimation) can be. Indeed, to our knowledge, very few papers have approached the uncertainty issue in GPR data (e.g., Barret et al., 2007; Lapazaran et al., 2016), even if uncertainty estimation is fundamental when a quantitative analysis is required as, for example, in

hydrological and climate change studies. In this work we computed step by step the value and the associated uncertainties of each parameter present in the equations, from the quantities effectively measured by each instrument to the final snow water equivalent values. In particular, because the uncertainties propagate through the calculations, we evaluated the "weight" of each uncertainty on the overall results. Moreover, to avoid any misunderstanding in the definition of measurement errors and uncertainties and to give a solid basis to uncertainty computation, in this work, we followed the NIST (National Institute of

Standard and Technology) guide lines on the expression of uncertainty in measurements (JCGM 100, 2008). Our approach allowed us to develop a procedure to accurately calibrate GPR in field conditions, that is, not only calibrate the radar system (which can be properly and accurately done in laboratory), but rather evaluate the overall performance accounting for systematic and random phenomena which can affect the field measurements.

## 2 Rationale of the measurement procedure

The goal of a measurement is to establish the numerical value of a quantity; however, as no measurement is completely free of uncertainty, an accurate evaluation of such uncertainty is part of the measurement process. The uncertainty should always be appropriately estimated and declared together with the results of the measurement, otherwise the reliability of the experiment could be questionable (Taylor, 1997; Kirkup and Frenkel, 2006). Furthermore, the consistency (or discrepancy) of independent measurements can only be proven, as will be discussed later, if the uncertainties associated to such quantities

are known.



Uncertainty estimation procedure has been debated for a long time among physicist and experimental scientists. When the data acquired during a laboratory and/or a field experiment is represented by a set of numbers, assuming that these data are the outcomes of repeated measurements, the uncertainty evaluation is straightforward, as it can be computed using statistical analysis (Taylor, 1997). However, there is a wide range of cases where the uncertainties cannot be estimated in such a way

and different procedures should be adopted. Quantitative geophysical studies, which are aimed at retrieving subsurface physical parameters (e.g., hydrogeophysics) are particularly affected by this type of problem; in addition, in field work it is not always easy to correctly evaluate the various contributions to the overall uncertainty of the retrieved physical parameter. The main goal of this study is to develop a strategy applicable to field data, to accurately estimate snow thickness (ST) and snow water equivalent using GPR. In theory, GPR is a self-consistent technique in snow thickness estimation as it is capable

to measure the signal two-way travel time (twt) and the wave velocity, if specific techniques are employed, like: Common-Mid Point (CMP) (Eisen et al., 2003; Bradford and Harper, 2006; Godio, 2009; Gustafsson et al., 2012), hyperbola fitting (Bradford et al., 2009), migration velocity analysis (Bradford and Harper, 2005; Holbrook et al., 2016) and reflected amplitude analysis (Forte et al., 2014). However, both two-way travel time and velocity parameters are subjected to uncertainties that are difficult to be evaluated, especially in field measurements, therefore they are usually neglected. In

particular, the uncertainty on two-way travel time measurement is strongly affected by the unknown "real" position of time zero on the GPR time scale and the unknown "real" shape and time length of the GPR signal after propagation and reflection (Economou and Kritikakis, 2016). It follows that it is not easy to properly pick the first break on transmitted and reflected signals and to compute the relevant uncertainties. On the other hand, the uncertainties on wave velocity measurements have been rarely evaluated (Barret et al., 2007).

In the present work, we collected four independent sets of data (TDR, density, snow thickness and GPR) in three different sites, according to the measurement chain sketched in Fig. 1. We acquired TDR and density data along the wall (vertical direction) of a snow pit and we estimated and compared the wave velocity computed with these two independent methods ("Pit measurements" square in Fig. 1). The consistency among the estimated velocity values, allowed us to apply a calibration procedure to compute the uncertainty on the GPR two-way travel time as follows. We selected the data acquired

with the hand probe (HP) for specific snow thicknesses, than using TDR velocity data and assuming a "theoretical GPR" generating and receiving only Dirac delta function signals we converted such data in two-way travel times ("predicted twt estimation" square in Fig. 1). We extracted the corresponding GPR data (i.e., collected in the same positions) and we estimated the uncertainties associated to the two-way travel time measurements under the assumption that, within the uncertainty interval, the wave velocity is constant in the three sites. Consequently, the variations in the two-way travel time

can only be ascribed to changes in snow thickness ("HP-GPR comparison" square in Fig. 1). Finally we estimated the depth of the snow and the associated uncertainties combining TDR and GPR data and, from that, the snow water equivalent. These values were compared with those computed using density data collected in the pit ("SWE estimation and comparison" square in Fig. 1).



All uncertainties associated with the snow physical parameters retrieved using this procedure, were calculated applying the most up to date approach to measurement uncertainty estimation, i.e., the "Guide to the expression of uncertainty in measurements" (JCGM 100, 2008).

The test site area is located in the Eastern Italian Alps, between Trentino-Alto Adige and Lombardia regions, in the Ortles-Cevedale group, the largest glacierized group of the Italian Alps (Carturan et al., 2013). In April 2014, we performed a field campaign in the nearby of the Solda glacier (coordinates 46°29′11″ N, 10°35′48″ E), at the base of the highest peaks of the Ortles-Cevedale group: Mt. Ortles (3905 m), Mt. Gran Zebrù (3851 m) and Mt. Cevedale (3769 m), which are aligned in a NW–SE direction; at the time of the survey, the snow cover had reached its maximum thickness, before starting to melt in the following month. The elevations of our test area were comprised between 2600 and 2800 m AMSL. We selected three different sites for our survey, which comprise two almost flat snow covered areas and the tongue of the Solda glacier, which was completely covered by snow during the period of the measurement campaign.

## 3 Methodology

### 3.1 Uncertainty estimation

For many years, in the scientific literature, the terms "measurement error" and "measurement uncertainty" have been used as synonyms. The "Guide to the expression of uncertainty in measurements" (GUM for short) (JCGM 100, 2008) clearly distinguishes these two terms, as described in Fig. 2 for the specific problem of snow thickness measurement. $ST_{true}$ is the true, but unknown, value of the snow thickness to be measured by any available technique; such quantity is named in GUM as the *measurand*. The value $ST_{meas}$ is the result of the measurement. The error in the evaluation of the snow thickness is unknown as well, as we do know neither the value of $ST_{true}$ nor all possible sources of errors (random or systematic) in the measurement. The best we can do is to estimate the value of the *measurand*, i.e., $ST_{meas}$, and compute the associated uncertainty $u$, taking into consideration all the possible contributions to such value. In this framework, $ST_{meas}$ represents the best estimate of the *measurand*, i.e., the snow thickness, and the uncertainty interval represents a range of possible values that are consistent with all observations and data and "…*that with varying degrees of credibility can be attributed to the measurand*" (JCGM 100, 2008).

The evaluation of a physical quantity and its associated uncertainty is more reliable, for obvious reasons, in a laboratory experiment; nevertheless the same rigorous approach can be applied in field measurements, if the uncertainty associated to each quantity accounts for all random and systematic effects that can be identified in the specific measurement procedure. For each measured parameter two possible approaches can be used (JCGM 100, 2008). i) If the quantity is the result of a series of repeated measurement, the best estimated value of such quantity ($ST_{meas}$ in our case) is the arithmetic mean of the values, and the uncertainty is computed as the standard deviation of mean, assuming that the data set follows a normal distribution. This procedure is named by GUM Type A evaluation of standard uncertainty. ii) If the quantity is not the result





of such repeated measurements (e.g., the thickness of the snow can only be measured once) the best estimate of the *measurand* is the single value $\text{ST}_{\text{meas}}$, and the associated uncertainty is computed on the basis of the available information assuming an *a priori* probability distribution from which the standard deviation can be calculated. This procedure is named by GUM Type B evaluation of standard uncertainty (for details see JCGM 100, 2008).

In this work, we mainly deal with Type B standard uncertainties as only in the procedure to evaluate the GPR two-way travel time uncertainty we performed a series of repeated measurements (Type A standard uncertainty). Indeed, for the uncertainty estimation of the quantities directly or indirectly measured in the snow, we have usually assumed an *a priori* uniform (i.e., rectangular) distribution. Such a choice is dictated by the lack of knowledge about the parameters controlling the measured quantities in the field. In fact, in the uniform distribution the probability density is zero everywhere except in a particular

region where the probability is constant, thus it is assumed that only the lower and upper bounds of the values are known and nothing can be said about the distribution inside this interval (Kirkup and Frenkel, 2006). The use of this type of distribution usually introduces in the calculation relatively large uncertainties but, on the other hand, prevents the underestimation of important effects on the uncertainty computation. For this distribution the standard uncertainty is $u = a/\sqrt{3}$, where $a$ is the half-width of the distribution. Note that in one case we have chosen a different distribution (normal distribution), for which

the appropriate standard uncertainty has been computed (see Sect. 4.3).

The standard uncertainties have been used to estimate the combined standard uncertainties $u_Q$ of the various quantities $Q$, under the assumption that the uncertainties $u_x, u_y, u_z, \ldots$ are independent and uncorrelated, applying the following equation:

$$u_Q = \sqrt{\left(\frac{\partial Q}{\partial x}\right)^2 u_x^2 + \left(\frac{\partial Q}{\partial y}\right)^2 u_y^2 + \left(\frac{\partial Q}{\partial z}\right)^2 u_z^2 + \cdots} \qquad (1)$$

Appendix A reports the values of all standard uncertainties associated to the measured quantities.

Finally, where appropriate, the estimated values of the uncertainties were compared considering two confidence levels, ~68 % and ~95 % that is, assuming infinite degrees of freedom, computing the expanded uncertainty $U = ku$ using a coverage factor $k = 1$ or $k = 2$ (JCGM 100, 2008).

### 3.2. Measurements in snow pits

### 3.2.1 TDR measurements

TDR measurements were acquired using a Tektronix 1502C Cable Tester connected to a three-prongs probe with conductors having a diameter of $(4.00 \pm 0.05)$ mm and a length $l = (30.0 \pm 0.3)$ cm. The distance between the central and the external conductor is $(3.20 \pm 0.03)$ cm and the theoretical characteristic impedance in air $(165 \pm 2)$ Ω, calculated according to Ball (2002). The measurements were performed inserting the probe horizontally in the pit wall at various depths. The depth of the





three pits was 2.94 m (site 1), 3.33 m (site 2) and 3.21 m (site 3) respectively, and matches in each site the overall thickness of the snow. In particular, in the first site only the top half of the pit was investigated (at irregular spacing) as some connection problems raised during the first day of data acquisition. In any case, in the second and third site, TDR measurements were regularly performed about every 30 cm to match the sampling interval of snow coring (see below), along the vertical length of the pit. The travel time $(t_2 - t_1)$ associated to the one-way signal propagation in the transmission line was estimated applying the derivative method (see e.g., Mattei et al., 2006) and the wave velocity was computed as:

$$v = \frac{l}{(t_2 - t_1)}, \tag{2}$$

The combined standard uncertainty on the velocity was computed using Eq. (1) and the Type B uncertainties on probe length and travel time ($u_l$, $u_{t_1}$ and $u_{t_2}$) estimated as described in Sect. 3.1 (uncertainty values reported in Table A1).

### 3.2.2 Density measurements

Snow density was measured sampling vertical cores nearby the edge of the snow pit, to obtain a continuous density profile vs. depth. Each sample was collected using a cylindrical corer, with a diameter $\phi = (4.700 \pm 0.029)$ cm and a height $d = (30.00 \pm 0.29)$ cm (i.e., a volume of about half liter). The mass $m$ of the samples was measured using a dynamometer. The combined standard uncertainty on the density was estimated using Eq. (1) and the standard uncertainties $u_\phi$, $u_m$ and $u_d$ associated to the mass of the snow and the height of the corer, respectively.

### 3.3 Measurements along profiles

### 3.3.1 GPR measurements

GPR profiles were collected along 7 transects for a total of 13 profiles, using a bistatic EkkoPro GPR system (Sensors and Software, Inc) equipped with 250, 500 and 1000 MHz antennas. GPR acquisition was performed in single fold mode with common offset, automatically collecting the data with an odometer at specific step size (5, 10 or 15 cm). A 4 traces stacking was set for all profiles and different time windows (from 70 to 300 ns) were chosen depending on the location of the profile. As reported in Table 1, all sections (aside from profile AB which has been investigated only with the 250 MHz antenna) have been acquired using 1000 MHz antennas. Profiles CD, OP and QR have also been investigated with the 250 MHz antennas and profiles GH, MN and IL with the 500 MHz antennas (see Table 1). Note that, among all profiles, only CD, GH and OP pass close to the snow pits.

The quality of the radar cross sections was very good for the entire data set. The interface snow/bedrock or snow/ice, as well as some internal layering in the snow pack, are well detectable in each section even without any gain applied, due to the very low signal attenuation of the snow. Figure 3 illustrates, as an example, the section acquired in the first site along profile AB with 250 MHz antennas.





### 3.3.2 Snow thickness measurements

Snow thickness was measured along each radar profile using an avalanche hand probe; the measurements were performed after radar data acquisition to measure the snow depth with the same degree of compaction (created by the weight of the antennas). The step size between consecutive measurements varied from 4 to 5.5 m depending on the profile. Snow thickness was ranging along the profiles from 1 to 4 m, depending on topography and snow cover, with an average thickness of about 2.50 m. As the hand probe has marks every 50 cm, we estimated a Type B uncertainty associated to each thickness measurement of 14 cm, that is, the standard deviation of a uniform distribution having 50 cm width (see Sect. 3.1). Furthermore, we estimated an uncertainty of 15 cm in the spatial position of the hand probe along the profile.

## 4 Analysis and Results

### 4.1 Wave velocity in snow

As discussed in Sect. 3.2.1, Eq. (2) allows to compute the value of the snow wave velocity using TDR measurements (probe length and travel time) as input parameters. A similar but more elaborate procedure can be applied to estimate the wave velocity starting from snow density values. Indeed, to reach this goal we have first to compute the relevant permittivities and then convert those into velocities. Such conversion is straightforward for the snow as it is non-magnetic and non-conductive, so the wave velocity depends mainly on the real part of permittivity. To compute the permittivity we applied two different relationships commonly used in dry snow studies (Kovacs et al., 1995; Godio, 2009; Previati et al., 2011; Di Paolo et al., 2015). The first one is the Looyenga mixing model between ice and air, which assumes a host material with spherical inclusions, and is given by (Looyenga, 1965):

$$\varepsilon_L = \left[ \frac{\rho}{\rho_{ice}} \left( \varepsilon_{ice}^{1/3} - 1 \right) + 1 \right]^3, \tag{3}$$

where $\rho$ is the snow density. We assumed $\rho_{ice} = (920 \pm 10)$ kg m$^{-3}$ (as reported by Kwok and Cunningham, 2008) and $\varepsilon_{ice} = 3.18 \pm 0.01$ (as measured by Bohleber et al. (2012) in a frequency range 10 MHz – 1.5 GHz).

The second one is Robin's empirical equation (Robin, 1975), given by:

$$\varepsilon_R = (1 + 8.45 \cdot 10^{-4} \rho)^2, \tag{4}$$

where $\rho$ is the snow density expressed in kg m$^{-3}$. The permittivities extracted from Eqs. (3) and (4) have been converted into velocity using the well-known equation:

$$v = \frac{c}{\sqrt{\varepsilon}} \tag{5}$$

where $c$ is the wave velocity in a vacuum. Again, the standard propagation formula (Eq. (1)) has been applied to estimate the relevant combined standard uncertainties on the permittivity values estimated using Looyenga and Robin models ($u_{\varepsilon_L}$ and $u_{\varepsilon_R}$), as well as those on the velocities ($u_{v_L}$ and $u_{v_R}$).





Figure 4 summarizes the results for the three sites. The velocity values extracted from TDR data are single points, blue dots with the associated uncertainty bars, and those computed from the density data are represented by solid segments (having the length of the corer, i.e., 30 cm). The values calculated from Robin's model are in red and those from Looyenga's model in black. The uncertainties associated to these values are represented by dash segments of the same color red and black,

respectively. Note that TDR data show the smallest uncertainties whereas those associated to Robin and Looyenga models are comparable. At site 1, as described in Sect. 3.2.1, the TDR data are only available for the first 1.50 m of snow cover, whereas in sites 2 and 3 the velocity values extracted from TDR measurements extend down almost to the bottom of the pit. All sites show the same trend: slightly higher velocity values near the surface, a decrease of velocity with depth up to 1.50 m and approximately constant values below such depth.

The velocity values shown in Fig. 4 have been used to compute a "weighted" average velocity of the electromagnetic waves in the entire snow pack. Figure 5 illustrates the procedure used to compute such quantity with TDR (a) and density (b) data. In particular, for the TDR data, which were acquired at depths $z_i$ inserting the probe horizontally in the pit wall, we assumed a model of constant interval velocities, that is, a model where the velocities $v_i$ are constant in layers of thickness $z_i^* = \frac{z_{i+1} \mp z_1}{2}$, with the sign minus (plus) valid for even (odd) values of $i$. Thus we calculated the average velocity as:

$$\langle v_{TDR} \rangle = \frac{H}{\sum_i z_i^*/v_i},$$ (6)

where $H$ is the snow pit depth (local snow thickness).

The combined uncertainty associated to the TDR average velocity was computed applying Eq. (1) with $u_H$, $u_{z_i^*}$ and $u_{v_i}$ estimated as Type B uncertainties. Note that due to the lack of data below 1.50 m, at site 1 we applied Eq. (6) assuming no velocity variation below the last TDR measurement point.

A similar equation has been applied to estimate the average velocity from density data:

$$\langle v_\rho \rangle = \frac{H}{\sum_i d_i/v_{\rho_i}}$$ (7)

where $d_i$ are the heights of the density samples (30 cm) and $v_{\rho_i}$ are the velocities computed with Eq. (5) for both Robin and Looyenga models. The combined uncertainty associated to these average velocities were computed as described in Sect. 3.1, with $u_H$, $u_{d_i}$ and $u_{v_{\rho_i}}$ estimated as Type B uncertainties (see Table A1). The results for the three sites are illustrated in Fig. 6

assuming two different confidence levels, that is, $k = 1$ and $k = 2$ as coverage factors. TDR values are in red, Robin in green and Looyenga in blue. In agreement with what already shown in Fig. 4, the measurements performed with TDR are the most accurate (about 1 % uncertainty) whereas for the velocity values computed from density data the uncertainties are slightly higher (about 2-3 %). Furthermore Fig. 6 shows that, taking into account the relevant uncertainties, no significant difference in wave velocity is observed among the three sites; this fact confirms that, in terms of electrical properties, the

snow pack was rather homogeneous. This condition allowed us to develop a procedure to estimate the overall uncertainty in the GPR two-way travel time and consequently, to accurately evaluate the uncertainty associated to snow thickness and snow water equivalent computed using radar data.



## 4.2 Two-way travel time uncertainty estimation

The overall uncertainty associated to the GPR two-way travel time is the sum of different uncertainties due to system characteristics, measurement procedure and data analysis. These effects are difficult to be separately evaluated, especially in field measurements, thus the best way to assess the uncertainty is to use a calibration procedure. We assumed that the hand

probe, which makes a direct measurement of the snow thickness, is the "best estimator" of such parameter and TDR the "best estimator" of wave velocity in the snow. In this way it is possible to compute the overall uncertainty accounting for different contributions like time base stability, antenna coupling, reflector properties and time picking procedure. To apply this method, we used the whole set of snow thickness data (i.e., collected on different profiles) to generate three subsets of measurements, one for each frequency, having the same number of samples. In particular, we have chosen, for each

frequency, the thickest value of the snow from which the same number of samples could be extracted. We found 7 hand probe readings at 300 cm depth (250 MHz), 230 cm depth (500 MHz) and 280 cm depth (1000 MHz) (see Table 2); all these snow thickness values were affected, as described in Sect. 3.3.2, by the same reading uncertainty of 14 cm. Then, assuming a "theoretical GPR" generating and receiving only Dirac delta functions (i.e., zero width signals) and using the average wave velocity estimated with TDR, we converted the hand probe thickness values in two-way travel time applying the following

equation (Lauro et al., 2013):

$$twt_{HP} = \sqrt{\frac{4}{v^2}\left(h^2 + \frac{s^2}{4}\right)} - \frac{s}{c} \tag{8}$$

Where $h$ is the snow thickness measured with the hand probe along the profiles, $s$ the antenna separation (38, 23 and 15 cm for the 250, 500 and 1000 MHz respectively) $v$ the average wave velocity measured with TDR in the specific site ($\langle v_{TDR} \rangle$). We also computed the combined uncertainty on $twt_{HP}$ using Eq. (1) and the Type B uncertainties $u_v$ and $u_h$ associated to

wave velocity and snow thickness (see Table 2).

Once defined for each frequency the set of seven "correct" two-way travel time values, we analyzed the radar sections to pick the corresponding traces. Considering that the spatial position of the hand probe is known with an uncertainty of ±15 cm, we extracted from each radar section all traces located inside this interval around the probe. Thus, depending on the step size of the radar profile, one, three or five traces were picked and averaged to obtain one radar trace for each hand probe

position. To compute the two-way travel time ($twt_{GPR}$) we applied a cross-correlation procedure to the GPR data (Lauro et al., 2013). For each radar section we defined a reference wavelet under the assumption that it represents the signal emitted by the antenna. The wavelet was extracted from a set of reflected signals collected at the snow/bedrock (first and second site) or snow/ice (third site) interface, in an area where the reflector was flat and sharp. Finally, we computed the cross-correlation between the reference wavelet and the extracted radar traces to pick the starting and ending point of the two-way travel time.

The procedure described above allowed us to compute three sets of predicted two-way travel time $twt_{HP}$ and to extract three sets of measured two-way travel time $twt_{GPR}$ related to snow layers of 300 cm, 230 cm and 280 cm, respectively. We treated these ($twt_{HP}, twt_{GPR}$) sets as the result of a repeatability (calibration) test where the $twt_{HP}$ quantities were assumed as



reference values and the $twt_{GPR}$ quantities were considered the data to be tested. Hence we estimated the combined uncertainty associated to the GPR two-way travel time as follows:

$$u_{twt_{GPR}} = \sqrt{u_{pick}^2 + u_{rep}^2} \qquad (9)$$

In Eq. (9) the component $u_{pick}$ (associated to the picking procedure) is a Type B uncertainty, estimated assuming a normal

distribution model, as the spectrum of the reference wavelet can be approximated to a Gaussian function having a RMS width $\sigma_{BW}$. That is:

$$u_{pick} = 2 \frac{1}{2\pi\sigma_{BW}} \qquad (10)$$

Conversely $u_{rep}$ (associated to the measurement repeatability) is a Type A uncertainty, which was computed using the set of seven calibration values as:

$$u_{rep} = \sqrt{\frac{1}{7(7-1)} \sum_{k=1}^{7} \left[ \left( twt_{HP_k} - twt_{GPR_k} \right) - \left( \overline{twt}_{HP} - \overline{twt}_{GPR} \right) \right]^2} \qquad (11)$$

Where $\overline{twt}_{HP} = \frac{1}{7}\sum_{k=1}^{7} twt_{HP}$ and $\overline{twt}_{GPR} = \frac{1}{7}\sum_{k=1}^{7} twt_{GPR}$. Note that Eq. (11) attributes to GPR the instability of the $twt_{HP}$ values computed using the "hand probe + TDR" data, thus it might contribute to an overestimation of the uncertainty $u_{twt_{GPR}}$. Table 2 summarizes the parameters used for the calibration and the estimated uncertainties for the three antenna frequencies together with the uncertainties associated to $twt_{HP}$. As shown by the values reported in the table, the component

$u_{pick}$ is always larger than $u_{rep}$, being about three times larger at 250 MHz and about twice at 500 and 1000 MHz. As a consequence, this component is the main contribution to the uncertainty on the GPR two-way travel time. Indeed the influence of $u_{rep}$ on the combined uncertainty (Eq. (9)) is practically negligible at 250 and 1000 MHz and it is rather small at 500 MHz. Furthermore, it is interesting to notice that the uncertainty $u_{twt_{HP}}$ on the predicted two-way travel time is quite large, being remarkably smaller than $u_{twt_{GPR}}$ only at 250 MHz and becoming even larger than $u_{twt_{GPR}}$ at 1000 MHz.

**4.3 Predicted vs. Measured two-way travel time**

Applying Eq. (8) to the entire dataset of hand probe values we converted depth into two-way travel time, we generated the scatter plots $twt_{GPR}$ vs. $twt_{HP}$ for the three frequencies and we estimated the parameters of the least squares linear fit for each frequency. Given the distribution of the profiles/antenna frequencies among the three sites (see Table 1), we obtained three scatter plots rather different in terms of spatial location of the data; in fact the 250 MHz plot was generated combining

site 1 and site 2 data, the 500 MHz only taking data from site 2 and the 1000 MHz combining data coming from all three sites. Furthermore, the data collected at 500 and 1000 MHz cover a larger two-way travel time range (10-40 ns) whereas those collected at 250 MHz are mainly distributed between 20 and 40 ns.

Figure 7 illustrates the scatter plots at the three frequencies, the fitting lines and the relevant parameters. All fits have been computed without imposing a zero intercept to check for possible biases. The plots indicate a fairly good linear trend for all





frequencies, with slope $a = 0.82 \pm 0.13$ at 250 MHz, $a = 1.038 \pm 0.083$ at 500, and $a = 1.023 \pm 0.048$ at 1000 MHz. Note that the uncertainty associated to the slope is the standard uncertainty of such quantity; that is, the slope has ~68 % chance of lying within the uncertainty (Kirkup, 1994). Regarding the intercept we found that the absolute values decrease when the frequency increases, being $b = (4.8 \pm 3.6)$ ns at 250 MHz, $b = (-1.0 \pm 1.7)$ ns at 500 MHz and

$b = (-0.2 \pm 1.2)$ ns at 1000 MHz, where the uncertainties associated to the intercepts have the same statistical meaning described above for the slope. Furthermore, to quantitatively evaluate the degree of correlation between N pairs of values, the correlation coefficient $r$ has to be associated to the distribution $P_c(r, N)$ which expresses the probability that the observed data could have come from an uncorrelated parent population (Bevington and Robinson, 2003). Small values of $P_c(r, N)$ imply that the observed variables are likely correlated. In our experiment we found the probabilities

$P_{c-250MHz}(r = 0.77, N = 105)$, $P_{c-500MHz}(r = 0.95, N = 735)$ and $P_{c-1000MHz}(r = 0.96, N = 157)$ always much lower than $10^{-3}$, indicating that all three $twt_{HP} - twt_{GPR}$ data sets exhibit a fairly good degree of correlation.

### 4.4 Snow water equivalent estimation

As a last step of the procedure, we computed the thickness of the snow and the snow water equivalent from GPR and TDR data as follows. We used the GPR data collected along CD, GH and OP profiles (i.e., the closest to the pits, see Table 1); as

the pits were approximately 1 m long, for each pit and for each frequency we extracted and averaged all traces present in the meter of profile located in front of the pit. The snow thickness $h_{GPR}$ was estimated as:

$$h_{GPR} = \sqrt{\frac{v^2}{4}\left(twt_{GPR} + \frac{s}{c}\right)^2 - \frac{s^2}{4}} \tag{12}$$

Where $v$ is the average velocity computed with TDR as described in Sect. 4.1. The uncertainty on this value was computed, as usual, applying Eq. (1), with $u_v$ and $u_{twt_{GPR}}$ the uncertainties associated to the average wave velocity (measured by the

TDR) and GPR two-way travel time and assuming a negligible uncertainty on $s$. Finally, the snow water equivalent was calculated using CRIM (Complex refractive Index Model) (Annan et al., 1994) as:

$$SWE_{CRIM} = 0.93 \left[\frac{\frac{c}{v} - 1}{\sqrt{\varepsilon_{ice}} - 1}\right] h_{GPR} \tag{13}$$

where the factor 0.93 accounts for the ice density reduction with respect to the water. The uncertainty $u_{SWE_{CRIM}}$ was computed combining the uncertainties $u_v$, $u_{\varepsilon_{ice}}$ and $u_{h_{GPR}}$ according to Eq. (1). The snow water equivalent values estimated

from GPR data (for the three sites) were compared with those computed for each pit, using the following equation:

$$SWE_{pit} = \sum_i d_i \rho_i, \tag{14}$$

where $d_i$ and $\rho_i$ are the height and the relative density of the $i$-th sample in the snow pit. The uncertainty $u_{SWE_{pit}}$ was computed combining the uncertainty $u_{d_i}$ and $u_{\rho_i}$, according to Eq. (1). Table (3) summarizes the results obtained using Eq. (13) and (14) together with some parameters useful to quantify the discrepancies between the methods. In particular, the





percentage difference between the estimated values of the snow water equivalent (not accounting for the uncertainties) has been computed as $\Delta = 100 \cdot |SWE_{GPR} - SWE_{pit}| / SWE_{pit}$ (to allow a comparison with other reference values) whereas to accounts for the uncertainties, the discrepancy has been evaluated according to the following formula:

$$|SWE_{GPR} - SWE_{pit}| \leq k \sqrt{u_{SWE_{GPR}}^2 + u_{SWE_{pit}}^2} \qquad (15)$$

where $k$ is the coverage factor. Note that in Table (3) the square root term of Eq. (15) is indicated as *df* (discrepancy factor). As shown in Table 3, the $\Delta$-parameter is of the order of 1-2 % for all profiles, except for the data collected on profile CD using 1000 MHz antennas, for which $\Delta = 6$ %. For this profile, there is also a weak discrepancy between the two snow water equivalent values when $k = 1$, whereas the other data do not present any significant difference between the snow water equivalent values, even for the smallest coverage factor.

**5 Discussion**

**5.1 Wave velocity comparison**

As one of the main goals of this work is to highlight potentials and limitations of different combinations of measuring techniques for snow thickness estimation, we start the discussion comparing TDR and density methods as wave velocity estimators. As described in Sect. 4.1, we computed the wave velocity using TDR and density data from measurements
collected along the wall (vertical direction) of three different snow pits excavated in the same area. As output of our analysis we reported, for each site, one set of values computed from TDR data and two sets of values estimated from density data (see Fig. 4). We actually used density data to test two different models (Robin and Looyenga) from which permittivity and, thus, wave velocity was calculated. As a consequence the two sets of values derived from density measurements cannot be considered totally independent results. In general, the velocities computed with the three methods (Eqs. 2-5) are in good
agreement as, considering the uncertainty intervals and the coverage factor $k = 1$, only few values do not overlap. However, the discrepancy among the values totally vanishes if a larger expanded uncertainty is assumed (i.e., choosing a coverage factor $k = 2$). The small point by point difference among the three data sets collected at each site (see Fig. 4) can be explained by the different sampling procedure used, as the TDR probe was inserted horizontally and the density probe vertically. On the other hand, the most significant differences between points (mainly present in the top portion of the snow
pit in site 2 and 3) could be due to the presence of some water in the snow, as the TDR estimates the overall velocity regardless the state of the snow, whereas Robin and Looyenga models are accurate only for dry snow. This statement is also supported by the observation that whenever the measurements do not agree, TDR velocity values are generally lower than those estimated with Robin and Looyenga models, as the presence of liquid water in the snow increases the bulk permittivity (decreases the wave velocity). This finding indicates that some caution should be taken when employing empirical models,
as they could introduce a bias in the velocity estimation due to their inability to take into account the presence of liquid water in the snow. The consistency among the three different methods applied to estimate the wave velocity is better seen when the



average velocity of the entire snowpack is computed, as described in Sect. 4.1. Indeed, assuming a larger expanded uncertainty (coverage factor $k = 2$) the uncertainty bars overlap nicely, as shown in Fig. 6; nevertheless a partial overlap is still preserved when $k = 1$. In this case, a small discrepancy is present only between the TDR average wave velocity computed for site 3 and those for site 1 and 2, whereas the velocity values estimated through density still agrees.

TDR measures the snowpack properties in a very similar fashion as GPR, thus it seems logical to rely on it for wave velocity estimation. Nevertheless our study indicates that there are other important reasons to consider this technique the "best estimator" of wave velocity: TDR is accurate (as will be better discussed later), very precise (only 1 % of uncertainty) and suitable for both dry and wet conditions. Therefore in our analyses we used TDR values (assuming $k = 1$) together with hand probe measurements to calibrate GPR data and estimate two-way travel time associated

uncertainties. However, our results also demonstrate that density methods can be successfully applied to compute wave velocity if the precision requested is not too high (i.e., the uncertainty needed is above few percent) and the snow is dry.

## 5.2 GPR travel time: calibration and uncertainties

The average wave velocity estimation allowed us to check the lateral homogeneity of the snowpack in the investigated area as, within the uncertainty interval, a similar velocity in the three sites was found. This condition is fundamental to correctly

apply a calibration procedure because in such procedure the compared quantities are two-way travel times predicted and physically measured for specific snow thickness values that are sparsely distributed in the entire investigated area. We applied a method (as described in Sect. 4.2) which is conceptually equivalent to a laboratory procedure used to check the performance of an apparatus by comparison with a calibrated instrument, assuming that the combination hand probe/TDR was the "best estimator" (i.e., the calibrated instrument) of the two-way travel time.

Before discussing the results of the calibration test it is important to remember that in the data analysis we dealt with combined uncertainties given by two or more uncertainty contributions. In estimating the magnitude of an uncertainty a convenient rule of thumb is to neglect all those terms that are less than 10 % of the largest uncertainty contribution (Bevington and Robinson, 2003). However, we decided to present the results keeping all contributions (see Sect. 4) as they can help to evaluate the weight of each uncertainty on the various quantities involved in snow thickness and snow water

equivalent estimation. We start analyzing the retrieved values of the uncertainty $u_{twt_{HP}}$ associated to the predicted two-way travel time; we found a constant value, 1.3 ns, for all frequencies (see Table 2) dominated by the uncertainty $u_h = 14$ cm on the snow thickness, being the uncertainty on the wave velocity rather small. In principle, such uncertainty can be reduced if a different procedure to measure the snow thickness is followed; for example, if the part of the probe sticking out the snow is measured with a meter or if a hand probe with a higher sensitivity (finer scale divisions) is used (Kinar and Pomeroy, 2015).

However, it should be bear in mind that large part of the uncertainty on snow thickness is due to the measurement of a very irregular surface; therefore the use of high sensitivity instruments would not significantly reduce the uncertainty. The uncertainty $u_{twt_{GPR}}$ on the GPR two-way travel time is given by the combination of two contributions (Eq. (9)): the first one,





$u_{pick}$, accounting for the time picking procedure and radar resolution and the second one which is purely random, $u_{rep}$, accounting for a combination of factors like system stability, antenna coupling, reflector properties, etc. We found that for all frequencies $u_{pick} > u_{rep}$ and $u_{twt_{GPR}} \approx u_{pick}$ (see Table 2); thus, in principle, $u_{rep}$ could be neglected without making a large mistake on $u_{twt_{GPR}}$. In fact, assuming that the uncertainty on the GPR two-way travel time is only due to the picking

procedure, the underestimation of the uncertainty is of the order of 3 % at 250 MHz, and 10 % for the 500 and 1000 MHz. However the computation of $u_{rep}$ provides important information about the overall stability of the measurement. Indeed, on the basis of our data (three sets of 7 data pairs) we found that $u_{rep}$ is always smaller than all the other uncertainties estimated in the calibration procedure (see Table 2), which implies a good repeatability in the GPR measurements, especially considering that the calibration has been performed on similar thicknesses but in different locations (i.e., different

measurement conditions). Regarding the magnitude of the uncertainties $u_{pick}$, it is no surprising that this is quite large, because the picking procedure we used to compute the two-way travel time is mainly affected by the radar bandwidth (Eq. (10)). Indeed, other ways to pick the signals could be chosen (e.g., Bradford, 2007), however the great advantage of the wavelet cross-correlation method is that it eliminates the ambiguity of the signal phase, which can be a large source of error in the two-way travel time estimation.

The calibration procedure allowed us to also make a direct comparison between $u_{twt_{HP}}$ and $u_{twt_{GPR}}$; we found that these uncertainties are of the same order of magnitude confirming that, in general, the calibration method chosen was correct as the apparatus under test and the calibrated instrument have the "same class of accuracy". However, from a direct comparison we also found $u_{twt_{GPR}} > u_{twt_{HP}}$ at 250 MHz, $u_{twt_{GPR}} \approx u_{twt_{HP}}$ at 500 MHz and $u_{twt_{GPR}} < u_{twt_{HP}}$ at 1000 MHz; this observation highlights that the highest antenna frequency is more precise than our two-way travel time "best estimator"

because at 1000 MHz $u_{pick} > u_{HP}$. It follows that, at this frequency, $h_{GPR}$ is more precise than $h$ which is not the case for the lower frequencies.

Once estimated the uncertainties associated to the GPR two-way travel time we were able to apply a regression analysis to the overall data set, subdividing the data by frequency. The computed parameters of the linear fit suggest that predicted and measured data are in very good agreement (see Fig. 7), being the correlation $twt_{GPR} - twt_{HP}$ highly significant for all three

frequencies. The data are well clustered around (and along) the regression line even if the plot at 250 MHz shows a slightly more sparse distribution of the data (i.e., a lower correlation coefficient). The value of the slope of the fitting lines at 500 MHz ($a = 1.038 \pm 0.083$) and 1000 MHz ($a = 1.023 \pm 0.048$) is unitary with a very small standard uncertainty and that at 250 MHz ($a = 0.82 \pm 0.13$) is close to 1 with a standard uncertainty one order of magnitude larger. This result demonstrates that TDR is very accurate in measuring the wave velocity in snow; in fact, keeping in mind that the plot at 250 MHz is made

with data collected in site 1 and 2, the one at 500 MHz with data only collected at site 2 and the plot at 1000 MHz with data collected in all three sites we can conclude that the velocity values predicted and measured are the same. Furthermore, the fact that the slope of the fitting line is the same at 500 and 1000 MHz also confirms that the velocity is essentially constant in the three sites. The small discrepancy in the linear fit slope between the 250 MHz data and those at higher frequencies, could





be explained by the low statistics at short two-way travel times as the low frequency data were collected (by chance) mostly where the snow was deeper (20-40 ns range). Regarding the intercepts (*b*-value), we can observe that those retrieved at 500 MHz and 1000 MHz are quite small, confirming that at these frequencies there is essentially no bias in the measured two-way travel times. As a consequence, for such data the estimation of $twt_{GPR}$ can be considered quite accurate. Conversely, the

intercept value at 250 MHz is significantly larger than those at higher frequency, highlighting the presence of a non-negligible bias. The reason of this discrepancy is not clear and should be investigated in more detail, even if it may be a consequence of the lack of data regularly distributed along the full range of the two-way travel time values. Something should also be said about the uncertainties on the b-values; in general, these are expected to be larger than the uncertainties on the $y_i$ values as the process to extrapolate the fitting line back to the y-axis can introduce large uncertainties (Taylor,

1997). As a consequence, the values we found for the three regression lines, and especially for the 250 MHz data, which are clustered at larger two-way travel times, can be considered quite reasonable.

**5.3 Snow water equivalent**

At the end of the overall procedure, once proven that $twt_{GPR}$ is reliable and accurate (especially at high frequency), we converted the $twt_{GPR}$ values in snow thickness, using the TDR average wave velocity, and we estimated the snow water

equivalent from the data collected on radar lines segments close to the three snow pits (as described in Sect. 4.4). Such data were compared with those retrieved from density measurements acquired in the pits, which we considered our snow water equivalent "best estimator". As described in Sect. 4.4 we evaluated the compatibility between measurements using both Δ-parameter, which has been used by other authors, and Eq. (15) that represents a more rigorous approach as it also accounts for the uncertainties. In general, we found an excellent agreement between GPR+TDR and density methods for all data

except in one case (as discussed below) (see Table 3). In fact, in our analysis Δ-parameter is of the order of 1-2 %, that is, similar or even better than those reported in previous works. For example, Sand and Bruland (1998) found 5 % at 500 MHz as maximum value for Δ-parameter, Lundberg et al. (2000) 5 % at 1200 MHz, Bradford and Harper (2006) 0.9 % at 900 MHz, and Bradford et al. (2009) 12 % at 900 MHz. Furthermore, according to Eq. (15), the data reported in Table 3 show that $SWE_{GPR}$ and $SWE_{pit}$ values (aside from one case) are always compatible even with $k = 1$. Regarding the data acquired

on profile CD with the 1000 MHz antennas, we found Δ= 6 % and a weak discrepancy between the two snow water equivalent values when the coverage factor is $k = 1$ (see Table 3). Such discrepancy was in some way unexpected as the other two measurements at 1000 MHz provided a good estimate of the snow water equivalent (see Table 3) and, more notably, the snow water equivalent value retrieved from 250 MHz data acquired on the same profile, is very accurate. Indeed, the analysis presented in this work suggests that the 1000 MHz measurements are generally more accurate and

precise than those performed with the low frequency antenna, thus the reason of the discrepancy should be sought into the measurement conditions. The source of this error is not clear however it may be in some way link to the antennas properties. In fact, as well known, the signal amplitude acquired by the GPR antenna is inversely proportional to the frequency; thus in





case of weak or irregular reflectors this may be a disadvantage as it affects the possibility to properly pick the reflected signal, introducing an error in the two-way travel time estimation.

## 6 Conclusions

In this work we studied the capability of GPR to properly estimate snow thickness and snow water equivalent when
supported by wave velocity TDR data. To reach this goal we performed a field calibration test to compare predicted and measured two-way travel times. We found that GPR is very accurate in estimating two-way travel times and such accuracy holds if the conversion into snow thickness is computed using TDR data. Snow water equivalent values computed using GPR data do not significantly differ from those calculated using traditional techniques, thus proving the reliability of the combined method. Furthermore, at high frequency the precision of GPR in estimating the snow water equivalent is similar to
that obtainable using density data.

The uncertainty budget evaluated for all quantities, measured and computed in this work, indicates that the most precise instrument is TDR and its impact on the uncertainty propagation is substantially negligible. On the other hand, the uncertainties associated to the GPR two-way travel time are relatively large and quite difficult to reduce, whereas the calibration test showed that, at least in the snow, GPR measurements are very repeatable.

The proposed approach, and in particular the calibration procedure, can be also applied in all those cases where the thickness of the snow is very variable and very few points with the same thickness can be found. In these conditions instead of using a repeatability test it is possible to build a calibration curve using predicted and measured data and compute all uncertainties according to the described procedure.

Finally our results demonstrate that when TDR is not available other valid alternatives can be found. GPR could be
combined with density measurements; in this case the accuracy of the retrieved parameters should be preserved but the precision would rapidly decrease due to the propagation of the uncertainties along the computation chain. Alternatively GPR data could be coupled with the hand probe measurements to compute snow wave velocity or to calibrate the velocity measured with GPR in specific locations, under the assumption that GPR and hand probe measurements are equally accurate, as proven in this work.

**Data availability**

The data used in this work are available and can be requested to the corresponding author.



## Appendix A

All standard uncertainties associated to the quantities used to evaluate the combined uncertainties are summarized in Table A1.

## Author contribution

F. Di Paolo, B. Cosciotti and E. Pettinelli carried out the GPR survey. S. E. Lauro and E. Mattei carried out the TDR survey. F. Di Paolo, M. Callegari, L. Carturan, R. Seppi and F. Zucca contributed to the excavation of the snow pits and carried out the measurements of density (in pits) and snow depth (along radar transects). F. Di Paolo performed the data analysis. E. Pettinelli and F. Di Paolo prepared the manuscript with contributions from all co-authors.

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



**List of figures**

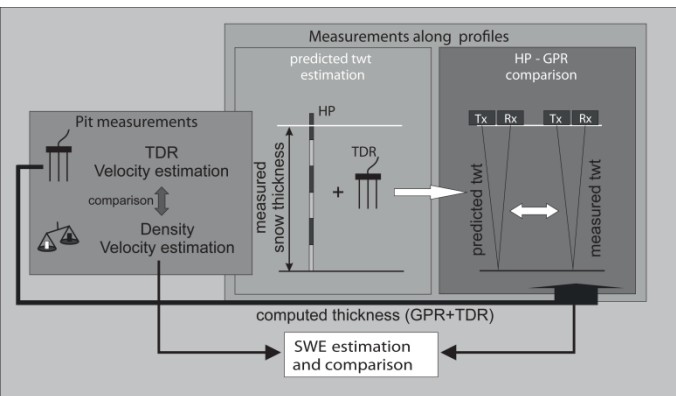

Figure 1: Schematic of the measurement procedure.

Figure 2: Error and uncertainty in snow thickness (ST) measurement according to GUM.

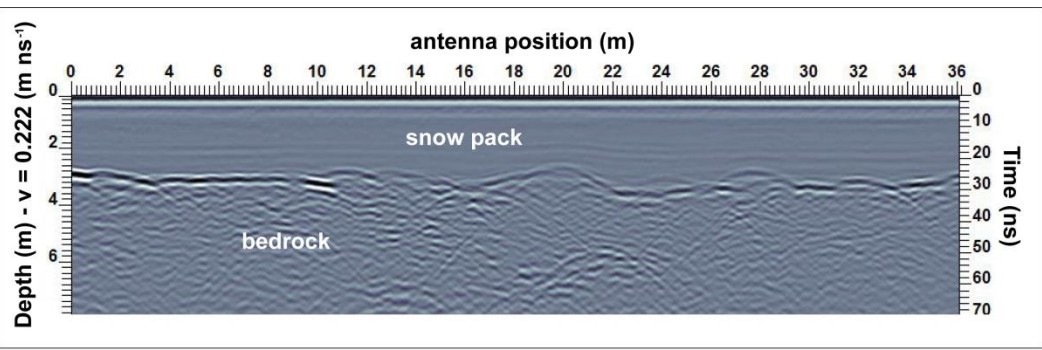

10  Figure 3: Radar cross section acquired using the 250 MHz antenna along trisect AB. The reflections from the snow pack/bedrock interface and the internal stratification in the snow, probably due to multiple snowfalls and wind effects, are quite evident.



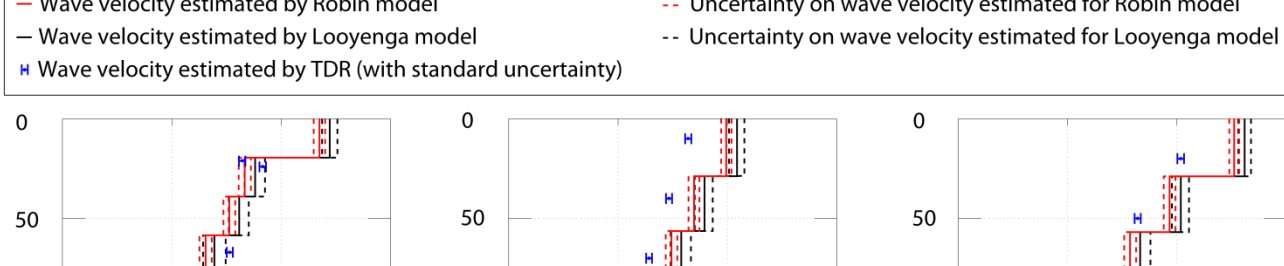

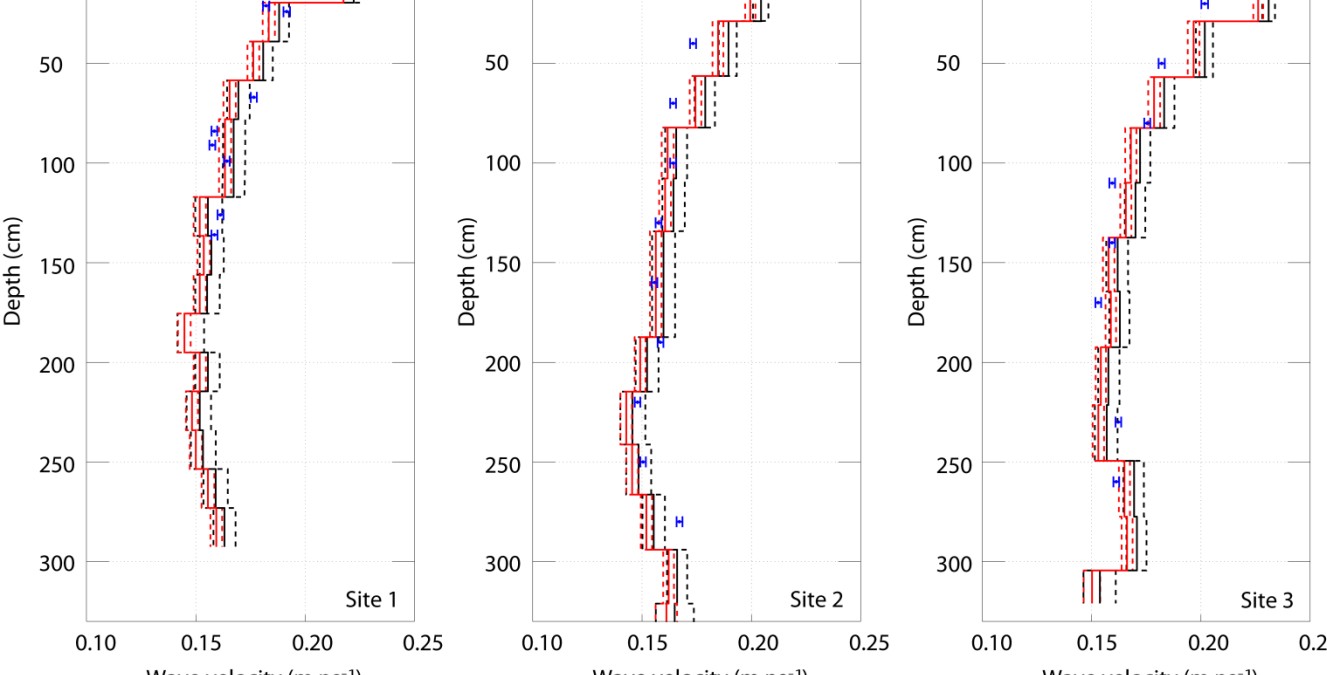

**Figure 4: Velocities estimated from TDR and density measurements on the wall of the three pits. Blue dots with the uncertainty bars are TDR values, red (Robin's model) and black (Looyenga's model) segments the values computed from density data. Dash segments indicate uncertainty intervals.**



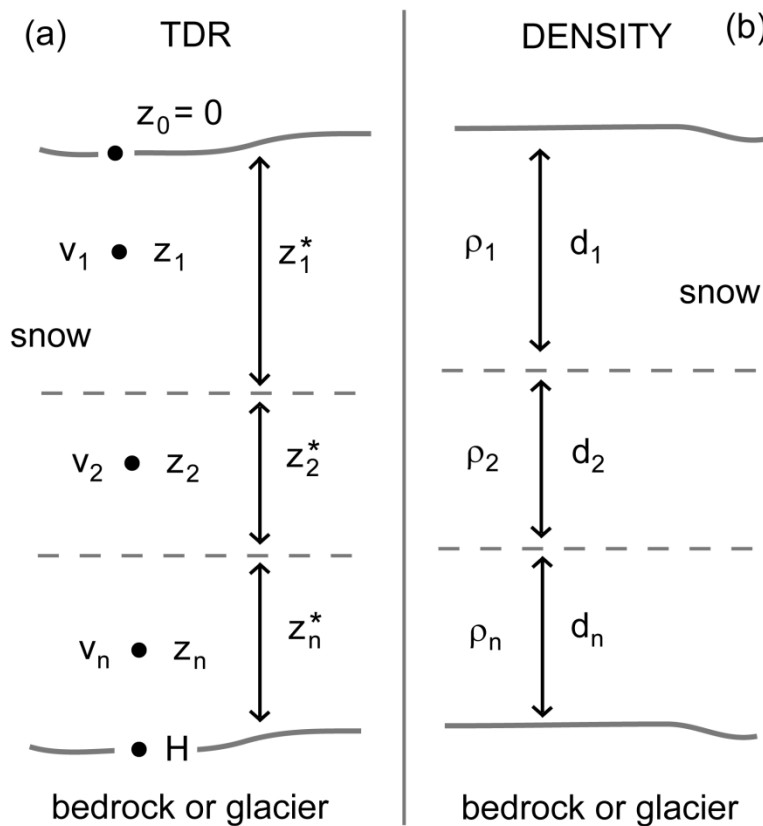

**Figure 5: (a) Sketch of the positions of the TDR probe $z_i$ and thickness of the snow layers $z_i^*$ used in Eq. (4). (b) Sketch of density sampling along vertical cores with height $d_i$.**



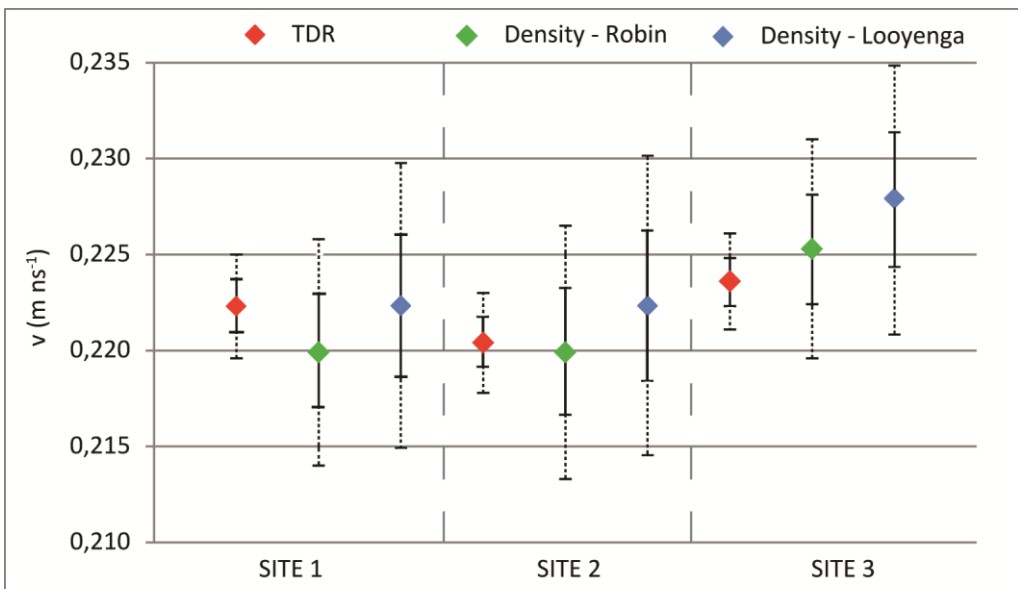

**Figure 6: Average wave velocity computed from TDR and density data in the different sites. Red dots are TDR values, green dots Robin and blue dots Looyenga values. Uncertainty bars are computed using a coverage factor $k = 1$ (solid lines) and $k = 2$ (dashed lines).**

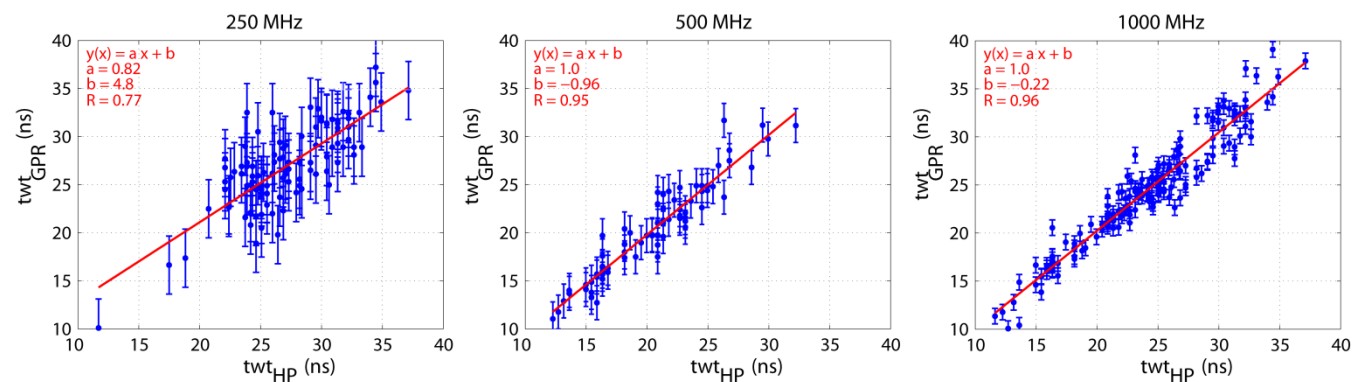

**Figure 7: Least squares linear fit for the three antennas frequencies and relevant fit parameters. Note the different two-way travel time ranges among the scatter plots.**


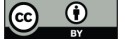

**List of tables**

**Table 1.** Site, length, antenna frequency and step size of the radar profiles.

\* Radar profiles passing close to the snow pits.

| Site | Profile | Length (m) | Frequency (MHz) | Step size (cm) |
|---|---|---|---|---|
| 1 | AB | 72 | 250 | 5 |
| 1 | CD* | 37 | 250, 1000 | 10, 5 |
| 2 | GH* | 90 | 500, 1000 | 5, 5 |
| 2 | IL | 85 | 500, 1000 | 5, 5 |
| 2 | MN | 85 | 500, 1000 | 5, 5 |
| 3 | OP* | 220 | 250, 1000 | 15, 5 |
| 3 | QR | 127 | 250, 1000 | 10, 10 |

**Table 2.** Calibration parameters and estimated uncertainties on $twt_{GPR}$ and $twt_{HP}$

| Frequency (MHz) | Snow thickness (cm) | Number of samples | $\sigma_{BW}$ (GHz) | $u_{pick}$ (ns) | $u_{rep}$ (ns) | $u_{twt_{GPR}}$ (ns) | $u_{twt_{HP}}$ (ns) |
|---|---|---|---|---|---|---|---|
| 250 | 300 | 7 | 0.11 | 2.9 | 0.87 | 3.0 | 1.3 |
| 500 | 230 | 7 | 0.21 | 1.5 | 0.87 | 1.8 | 1.3 |
| 1000 | 280 | 7 | 0.43 | 0.74 | 0.32 | 0.81 | 1.3 |





**Table 3.** Snow water equivalent measured in snow pits and estimated from GPR data using CRIM.

| Site | Radar profile | Frequency (MHz) | $SWE_{GPR} \pm u_{SWE_{GPR}}$ (mm) | $SWE_{pit} \pm u_{SWE_{pit}}$ (mm) | $\Delta$ | $\left\|SWE_{GPR} - SWE_{pit}\right\|$ (mm) | $kdf$ $k = 1$ | $kdf$ $k = 2$ |
|---|---|---|---|---|---|---|---|---|
| 1 | CD | 250 | $1266 \pm 136$ | $1267 \pm 50$ | <1 | 1 | 145 | 290 |
|   |    | 1000 | $1185 \pm 43$ |  | 6 | 82 | 66 | 132 |
| 2 | GH | 500 | $1408 \pm 82$ | $1435 \pm 50$ | 2 | 27 | 96 | 192 |
|   |    | 1000 | $1400 \pm 46$ |  | 2 | 35 | 68 | 136 |
| 3 | OP | 250 | $1289 \pm 134$ | $1260 \pm 47$ | 2 | 29 | 142 | 284 |
|   |    | 1000 | $1271 \pm 43$ |  | 1 | 11 | 64 | 127 |

5   **Table A1.** Standard uncertainties associated to the measured variables. Considering a uniform distribution for all the variables: $p = \sqrt{3}$.

| Uncertainty | Value | Standard uncertainty |
|---|---|---|
| $u_t$ | $0.002/p$ ns | 0.0012 ns |
| $u_l$ | $0.05/p$ cm | 0.029 cm |
| $u_\phi$ | $0.05/p$ cm | 0.029 cm |
| $u_m$ | $5/p$ g | 2.9 g |
| $u_d$ | $0.5/p$ cm | 0.29 cm |
| $u_H$ | $1/p$ cm | 0.58 cm |
| $u_z$ | $0.5/p$ cm | 0.29 cm |
| $u_h$ | $25/p$ cm | 14 cm |