# Peer review of "Uncertainty budget in snow thickness and snow water equivalent estimation using GPR and TDR techniques"

_The Cryosphere, 2016_

## Referee Comment (RC1) · Anonymous Referee #1 · 7 Feb 2017

Based on the comments and remarks within the supplemented PDF my suggestions is to reject the manuscript in the present form unless inconsistencies can be unambiguously resolved, the data interpretation revised and the discussion of the results be set into a more general valid, broader context.

Please also note the supplement to this comment:
http://www.the-cryosphere-discuss.net/tc-2016-267/tc-2016-267-RC1-supplement.pdf

[Figure]

**Supplement:**

Review on tc-2016-267: Uncertainty budget in snow thickness and snow water equivalent estimation using GPR and TDR techniques

**Summary**

The authors present an analysis on ground-penetrating radar (GPR) and timedomain reflectometry (TDR) measurements for estimating the snow height and snow water equivalent. The main focus of the manuscript is the comparison between a combination the two techniques and the traditional sampling method and on determining the various sources of error according to the "Guide to the expression of uncertainty in measurements". All measurements were performed during April 2014 in the Italian Alps in the vicinity of the Ortles-Cevedale range. The authors measured at three different sites on an elevation of 2600-2800 m. Aspect and slope angle were not reported. On these three sites the authors performed several GPR transects with three different antennas (250 MHz, 500 MHz and 1000 MHz) in single fold mode with common offset. Snow depth was measured with an avalanche probe with 0.5 m resolution. In addition, they performed classical volume-mass density samples with 0.3 m vertical resolution and TDR measurements (length of probe = 0.3m) within three snow pits in the vicinity of the GPR transects. Using the mixing model of Looyenga and the empirical equation by Robin the authors calculate the bulk permittivity and consequently the wave velocity and then snow height and snow water equivalent (SWE), do an inter-comparison and try to estimate the various influence of present sources of uncertainty.

**Evaluation**

The presented and analyzed approach is promising, but not novel. Increase of general valid knowledge may be given due to the presented methods on how to assess uncertainty – even though some approaches and intentions by the authors are questionable (e.g. 0.5m resolution of avalanche probe). Also by using the combination of TDR and GPR, some potential for novelty may be found, but the authors do not fully exploit the possibilities of this sensor combination. As explained in more detail below, I believe that the authors made some severe mistakes in (1) their concept of measurements, (2) data analysis and (3) approach of how to estimate the various uncertainties. In addition, presented results seem for some reasons to be inconsistent. Most severe inconsistencies are between Fig. 4 and Fig. 6

Further, the authors miss to put their analysis into a broader context. It seems that they are not aware of recent publications on very related topics such as the combination of GPR and GPS sensors to derive SWE (Schmid et al., 2015) or more general analysis on determining snow properties with radar technology and its limitations (Heilig et al., 2014; Okorn et al., 2014; Schmid et al., 2014). The authors do refer to the publications by Bradford and Harper (2006); Bradford et al. (2009), but miss to draw the correct conclusions or discuss them into a broader context. In fact, the discussion is poor, inconsistent and sometimes wrong; consequently it needs considerable improvement.

The language is often not concise, does often not use or even miss-uses broadly accepted terminology within the snow science community. In addition, large parts of the manuscript need grammatical improvements and must be checked by a native speaker.

Based on the above and the detailed general remarks my suggestions is to reject the manuscript in the present form unless inconsistencies can be unambiguously resolved, the data interpretation revised and the discussion of the results be set into a more general valid, broader context.

**General remarks**

There are several issues or misunderstandings within the manuscript:

**$\Rightarrow$ Wave velocity values in Figure 4**

Wave velocity values presented in Figure 4 are mostly below 0.2 m/ns for all three snow pit sites. TDR based velocity calculations have even lower values for the top 0.7m of the test sites 2 and 3. The values presented in Figure 4 correspond either to very dense dry snow (more than 500 kg/m3) or to wetsnow conditions with a dry snow density of around 350-400 kg/m3 and an average volumetric liquid water content of 0.02 to 0.04 (see Fig. 1 within the review). When I recalculated the wave velocity on the SWE values presented in Table 3 and the values for snow depth at the profiles given on page 7. I obtain densities of 390-430 kgm-3 which with the Eq by Robin and Looyenga result into wave velocities of  $\sim 0.22$  m/ns. I cannot reproduce these average values based on Figure 4 in your manuscript. The values correspond better to Figure 6 in your manuscript. If, however, the values of Figure 4 are the correct ones, the conclusions are that your snow was wet. As a consequence your simple mixing model is not applicable and has to be replaced by more sophisticated mixing models (Denoth, 1980; Denoth, 1994; Roth et al., 1990). This will influence the TDR measurements and consequently the used approach of calibrating your GPR data and calculating SWE are not valid and thus large parts of the manuscript cannot be published.

Wave velocity (m/ns)

Fig. 1: Wave velocity in as function of volumetric liquid water content and different dry-snow densities. Red bar with whiskers show range of values given in Figures 4 and 6 of the manuscript.

**⇒ Inconsistencies between wave velocity values in Figure 4, Figure 6 and average values of Table 3**

As mentioned several times in the specific remarks and comments within the text, your values of Figure 4 and Figure 6 are not consistent. If I calculated density-based values of wave velocity calculated with the Robin model for site 2, I obtain an average weighted wave velocity of 0.16 m/ns which corresponds to densities of ice, or very wet snow. In fact, the values of Figure 6 correspond more or less to the values given in Table 3. Based on your snow heights and SWE values, I recalculated the bulk snow density to 390-430 kg/m3 which corresponds to a wave velocity of 0.22 which is given in Figure 6. Please comment on the inconsistencies!

**$\Rightarrow$ Concept of measurement**

Unfortunately you made several severe mistakes while sampling:

- The use of a 0.5m resolution for your hand-probed snow height ruins the entire analysis.
- You sample density with a fixed spacing instead of using snow layers with different hand or ram hardness and thus different densities
- Consequently your TDR measurements may not represent the cylinder sample which complicates the search for uncertainties or errors. When only little amount of water is present, we will not identify whether the differences in Fig.4 are caused by water or higher values of density.
- I believe that your analysis of the TDR travel time is inaccurate. Please provide more details how you processed the data including figures.

**$\Rightarrow$ Concept of estimating uncertainty**

It remains unclear, why the authors have chosen in some cases to use their definition of Type B uncertainties, while is would have been possible to stay with a Type A data uncertainty by e.g. sampling density or snow height several times in close proximity.

In addition, I have the feeling that Type B uncertainties are purely driven by the units given by the chosen estimated values. E.g. with TDR small differences in estimating t2 may result in large differences for the permittivity. I want to see a broader discussion on that topic.

**Specific remarks**

For the more specific remarks, please check the direct comments within the text.

[revised manuscript text omitted]

**10 3.2.2 Den**

---

## Author Comment (AC1) · 17 Feb 2017

tc-2016-267-RC1 – Comment to Anonymous Referee #1 Interactive comment on "Uncertainty budget in snow thickness and snow water equivalent estimation using GPR and TDR techniques" by Federico Di Paolo et al.

We would like to apologize with the Associated Editor and the reviewer 1 about Fig.4 which, as highlighted by the referee, is wrong because the X-scale is incorrect due to a mistake in the code when it generated the picture. This mistake has been cause of confusion. The following Figure is the correct one and, as it can be seen, is in agreement with Figure 6 and the overall results presented in the original manuscript.

[Figure]

Please also note the supplement to this comment:
http://www.the-cryosphere-discuss.net/tc-2016-267/tc-2016-267-AC1-supplement.pdf

[Figure]

Figure 4 plots: Wave velocity (m ns⁻¹) vs Depth (cm) for Site 1, Site 2, Site 3.

Legend:
- Wave velocity estimated by Robin model
- Wave velocity estimated by Looyenga model
- Wave velocity estimated by TDR (with standard uncertainty)
- Uncertainty on wave velocity estimated for Robin model
- Uncertainty on wave velocity estimated for Looyenga model

**Fig. 1.** Figure 4

---

## Author Comment (AC2) · 21 Feb 2017

We would like to apologize to the Associated Editor and Reviewer 1 regarding Fig.4 which, as highlighted by the referee, had an incorrect X-scale due to a mistake made by the code when it generated the picture. This mistake has been cause of confusion. The following Figure is the correct one and, as can be seen, it is in agreement with Figure 6 and the overall results presented in the original manuscript.

Furthermore, we wish to thank the referee for his/her efforts, even though we strongly disagree with the overall assessment and with several specific comments, as discussed below. The reviewer's comments are given in red, followed by our replies in black.

[Figure]

Fig.4 – corrected version

⇒ Wave velocity values in Figure 4

Wave velocity values presented in Figure 4 are mostly below 0.2 m/ns for all three snow pit sites. TDR based velocity calculations have even lower values for the top 0.7m of the test sites 2 and 3. The values presented in Figure 4 correspond either to very dense dry snow (more than 500 kg/m3) or to wetsnow conditions with a dry snow density of around 350-400 kg/m3 and an average volumetric liquid water content of 0.02 to 0.04 (see Fig. 1 within the review). When I recalculated the wave velocity on the SWE values presented in Table 3 and the values for snow depth at the profiles given on page 7, I obtain densities of 390-430 kgm-3 which with the Eq by Robin and Looyenga result into wave velocities of ~ 0.22 m/ns. I cannot reproduce these average values based on Figure 4 in your manuscript. The values correspond better to Figure 6 in your manuscript. If, however, the values of Figure 4 are the correct ones, the conclusions are that your snow was wet. As a consequence your simple mixing model is not applicable and has to be replaced by more sophisticated mixing models (Denoth, 1980; Denoth, 1994; Roth et al., 1990). This will influence the TDR measurements and consequently the used approach of calibrating your GPR data and calculating SWE are not valid and thus large parts of the manuscript cannot be published.

**Reply**: As mentioned above the figure was incorrect, thus the comments of the reviewer on this specific issue do not apply to our field conditions. In fact, from the correct values of the wave velocity we can infer that the snow is essentially dry. As requested by Referee 1, using the formula reported in *Denoth, 1994* the

snow water content is always lower than 1%, as is clear comparing the plot made by the referee with the one of the density reported below.

[Figure]

**⇒ Concept of measurement**

**Unfortunately you made several severe mistakes while sampling:**

**• The use of a 0.5m resolution for your hand-probed snow height ruins the entire analysis.**

**⇒ Concept of estimating uncertainty**

**It remains unclear, why the authors have chosen in some cases to use their definition of Type B uncertainties, while is would have been possible to stay with a Type A data uncertainty by e.g. sampling density or snow height several times in close proximity.**

**Ref. comment (page 18, line 26): This fact reduces your whole discussion on uncertainty to absurdity. The uncertainty of your probe reading is too large from the very beginning.**

**Ref. comment (page 18, line 26): I totally disagree! If you were right, all laser-scan based measurements would be useless! If you really believe in this arguement, you must proof it with literature or examples.**

**Ref. comment (page 12, line 6): This is a very weak point in your analysis. The measured snow height with the probe somehow represents your target variable and has the lowest resolution. Why did you not combine the 0.5m resolution with a standard ruler so that you end up with a cm resolution?**

**Ref. comment (page 10, line 1): not true! Depends on your method! In addition it totally depends on the used scale! If you assume that the snow height on almost flat terrain, you can easily compute 10 probe measurements in an array of 10 by 10 cm which will result in a Type A evaluation of your uncertainty, if you you define a 100cm^2 foot print, which should be fine for your radar beam. Your Type B is again totally driven by the scale of your measurement device, in your case by the very, very coarse setting (0.5m) of your probe.**

**Reply**: We did not introduce any "personal" method to compute the uncertainties but we followed the most up to date procedure. See for example: http://www.bipm.org or https://www.nist.gov/. However, in the following we better explain why our procedure is rigorous and correct and why using a hand probe with a much higher resolution scale would not appreciably improve our analysis.

According to GUM, the overall uncertainty on each quantity should be computed considering all possible sources of uncertainties, that is Type A + Type B to estimate the so called uncertainty budget. In the paper, for the hand probe measurements, we have applied a simplified procedure estimating the overall uncertainty using the standard deviation of a uniform distribution (type B uncertainty). In the following we demonstrate that the chosen value of uncertainty is only 1.5 times the uncertainty computed using a "more refined" scale on the hand probe (e.g., 2 cm scale ruler) but taking into account other possible sources of uncertainty. The computation has been performed for the 250 MHz antennas, but can be extended to the other antennas.

[Figure]

We first computed the footprint of the antenna on the bedrock for a snow height of 300 cm (see Fig.R1a and R1b). For each measurement the footprint is $(576 \times 288) cm^2$ (i.e., much larger than 10x10 cm - *compare to Ref. comment: Page 10, line1*). In such a large area, and considering the site (alpine region), we can assume bedrock surface irregularities of at least 20 cm, so in the antenna footprint the snow height interval is 290 - 310 cm (see Fig.2R). Furthermore, for simplicity we will neglect the undulation of the snow.

We consider now some of the possible source of errors that can affect our measurements (see for example: A Beginner's Guide to Uncertainty of Measurement, by Stephanie Bell –National Physical Laboratories, 1999):

**a) List of possible errors due to hand probe**
1) Is the hand probe calibrated? What is the uncertainty on calibration?
2) Has the hand probe been modified by the use?
3) What is the hand probe resolution?

**b) List of possible errors associated to snow height**
1) Are the top and the bottom of the snow layer well defined and constant?

**c) List of possible errors due to measuring process**
1) How well do the hand probe starting points (i.e., the tip at the bottom) coincide with the bottom of the snow layer?
2) Is the rod perpendicular to the snow surface at the measurement point (rod parallel to the snow height)?
3) How repeatable are the measurements?

In our field conditions, as these questions are not easy to quantify, we will simplify our computation as follows, bearing in mind that such simplification will certainly introduce an underestimation of the overall uncertainty.

We assume to measure the snow with a hand probe having marks every 2 cm. Thus the uncertainty on reading (Type B uncertainty assuming a uniform distribution) is:

$$2a = 2cm$$

$$\sigma_{read} = \frac{a}{\sqrt{3}} = 0.6cm$$

Now, we randomly generated 10 measurements of the snow height (according to the range of height variability given above that is ± 10 cm). The measurements are:

300, 302, 310, 295, 308, 291, 298, 304, 290, 305

Note that these readings are taken assuming a resolution of 1 cm (half of the marks distance on the scale). From these data we can estimate the Type A uncertainty averaging the measurements and the standard deviation uncertainties:

$$\overline{x} = 300.3cm$$

$$\sigma_x = 6.8cm$$

$$\sigma_{\overline{x}} = 2.2cm$$

From this calculation we see that the uncertainty associated with a single measurement (and any other $n+1, n+2, n+...$ measurements that can be made with the hand probe) is about 7 cm, that is over 3 times the divisions on our hypothetical hand probe scale (i.e., 2 cm). This means that we are using a "system" that is too refined for our measurement and the uncertainty is much larger than the scale division. In other words, the spread of the data due to bedrock (or bottom layer) irregularities is much larger than the sensitivity of the instrument.

Furthermore, we can try to compute the bias error associated with the inclination of the probe (see Fig.R2). If we assume that we insert the probe with an average angle of 10° with respect to the vertical and we also assume that the uncertainty on this inclination is 10° (between 15°and 5°) we can compute for 300 cm snow height a bias error (BE) as follows:

$$c = \frac{b}{\cos\theta} = 304.6cm$$
$$BE = 4.6cm$$

The uncertainty on this error is:

$$c' = \frac{b}{\cos\theta'} = 310.6cm$$

$$c'' = \frac{b}{\cos\theta''} = 301.1cm$$
$$\Delta c = 9.5cm$$

Now assuming a uniform distribution, the type B uncertainty for the inclination is:

$$\sigma_{incl} = \frac{9.5}{2\sqrt{3}} = 2.7cm$$

As the bias error is, by definition, always there and overestimates the snow height, it should be subtracted from the average value:

$$SH_{corr} = 300.3 - 4.6 = 295.7 cm$$

The combined standard uncertainty associated to this value is:

$$\sqrt{\sigma_{\bar{x}}^2 + \sigma_{read}^2 + \sigma_{incl}^2} = \sqrt{2.2^2 + 0.6^2 + 2.7^2} = 3.5 cm$$

This standard uncertainty represents 68% of probability that the measurement falls between ±3.5cm. If we use a coverage factor, e.g., k=2 (as suggested by GUM), we will have that about 95% of our measurements will fall in the interval ±7.1 cm.

We can conclude that the uncertainty on our measurement is at best about 7 cm. We should remember however that we are underestimating the uncertainty, as several items in points *a)*, *b)* and *c)* above weren't considered because they are too difficult to estimate in the present case. For example, it is not easy to estimate the effect on snow height measurements of rocks or ice lenses present in the snowpack. As a consequence is quite possible that, in our case, the overall uncertainty is ≥10 cm.

In measurement theory (and practice) the best choice of instrument scale division is driven by the overall (or maximum) uncertainty, thus to optimize our calibration in our specific field conditions, we should have used a hand probe with marks divisions computed using the combined standard uncertainty computed above (i.e., 10 cm). From this value we can compute the suitable scale division assuming a uniform distribution as follows: $2a = 2 \times 10 \times \sqrt{3} = 34.6 cm$. In our work we used a probe with 50 cm distance between the marks, so in analogy with the example discussed above, with a resolution of 25 cm, which for a uniform distribution gives a type B uncertainty of 14 cm. This is an uncertainty of about 1.5 times the value we have found using a hand probe with only 2 cm marks distance.

If we use for the hand probe readings a uncertainty of 10 cm (assuming that, as we will demonstrate below, the uncertainties on the TDR are correct) we found a $u_{twt_{HP}} = 0.91 ns$ which does not change the results presented in the original manuscript (cf. Table 2 in the manuscript).

We can conclude that, in the specific field conditions of our study, the use of a more precise technique like laser, which samples a very small area [see e.g., Lee et al., 2015], would not significantly improve the GPR calibration due to all sources of errors present in the large radar footprint.

⇒ **Concept of measurement**

**Unfortunately you made several severe mistakes while sampling:**

**• I believe that your analysis of the TDR travel time is inaccurate. Please provide more details how you processed the data including figures.**

⇒ **Concept of estimating uncertainty**

**In addition, I have the feeling that Type B uncertainties are purely driven by the units given by the chosen estimated values. E.g. with TDR small differences in estimating t2 may result in large differences for the permittivity. I want to see a broader discussion on that topic.**

**Ref. comment (page 11, line 6): Provide a Figure on the signal and explain exactly how you picked the derivate. Since snow tends to give a weak signal, the slope indicating the end of your sensor tends to be flat which introduces again a source of uncertainty, since you have to pick a certain time of your**

**signal. Please comment on that and make statements on how you could eventually prevent this source of error for future investigations.**

**Ref. comment (page 18, line 7): In my opinion not shown and biased by your chosen parameters for u.**

**Reply**: The comments made by the referee require some explanation about TDR working principle and its applicability to snow measurements. In the following we summarize such information, including an example of the analysis of a waveform collected during the measurement campaign. Time domain reflectometry (TDR) is a valuable method for measuring the electromagnetic (e.m.) wave propagation velocity in solid, granular and liquid materials. Our TDR system consists of an open ended three-prong line (hereafter referred to as probe line) that can be embedded in the material; the instrument generates a stepwise signal with a 200ps rise time [Tektronix 1502 cable tester]. The output of the TDR system consists essentially of the signal recorded at the feeder line input, which displays the results of the reflection processes taking place in the system. A first partial reflection occurs when the wave front from the generator comes across the impedance discontinuity at the probe line input. In fact, the probe line is designed in a way to exhibit an impedance mismatch at the generator feeder line. A second reflection takes place at the end of the probe, with a unit reflection coefficient as the probe is open-ended. The wave velocity is estimated from the signal two-way travel time; a complete discussion about the multiple reflections is given by *Yanuka et al.* (1988) and *Topp et al.* (1988). An example of the TDR trace acquired during the survey in the Ortles-Cevedale group is reported in Fig. R3. The first and second arrows indicate the reflection at the beginning and the end of the probe line. As we can see, the two reflections are very clear and can be used to determine the two way travel time of the signal along the TDR probe: in fact, the two way travel time appears in the TDR trace as the time interval between the reflections at the probe edges. For non-magnetic materials (like snow), this time is given by $\Delta t = 2L\sqrt{\varepsilon_a}/c$, where $L$ is the length of the probe, $c$ the light speed in a vacuum, and $\varepsilon_a$ is the apparent relative permittivity given by [*Von Hippel*, 1994]

$$\varepsilon_a = \varepsilon' \frac{(1+\sqrt{1+\varepsilon''^2/\varepsilon'^2})}{2},$$

where $\varepsilon'$ and $\varepsilon''$ are the real and imaginary part of the permittivity. Therefore, the two way travel time can be used to compute the apparent permittivity or, alternatively, the velocity of the em signal in the medium under test (i.e., snow), by using:

$$\varepsilon_a = (c\Delta t/2L)^2$$

$$v = 2L/\Delta t$$

Note that $\varepsilon_a$ reduces to the real part of permittivity if losses are negligible.

[Figure]

[Figure]

Fig. R3 Left panel represents the TDR response; right panel represents the derivative of the TDR response.

To pick the t1 and t2 times (which correspond to the beginning and the end of the probe, respectevely), we used the derivative method which is widely discussed in Robinson et al. (2003), Robinson et al. (2005) and Mattei et al. (2006) and is briefly described in the following. An example of the time derivative of the TDR response is reported in the right panel of Fig. R3: the time derivative of the TDR response can be modelled by a series of Gaussian functions

$$\dot{r}(t) = \sum_{j=1}^{n} A_j \exp[(t - t_j) / \sigma_j]^2 ,$$

where the index $j$ runs over the successive reflections present in the TDR trace. We applied a minimization procedure to the waveform shown in Fig.R3 /right panel) up to $j=2$ to evaluate the values of $\Delta t$ from the time difference $t_{j+1} - t_j$.

We use the time sampling of the Tektronix cable tester (0.0041ns) to estimate the uncertainty on each $t_j$ value assuming a uniform distribution (Type B uncertainty).

Furthermore, we would like to reply to this comment: **"In addition, I have the feeling that Type B uncertainties are purely driven by the units given by the chosen estimated values. E.g. with TDR small differences in estimating t2 may result in large differences for the permittivity. I want to see a broader discussion on that topic".**

As shown by the waveform (see Fig.3R), the measurement of t2 is well constrained, being an open-ended probe, that is the impedance is Z=∞ and the reflection coefficient Γ=1. As a consequence, the effect of t2 on the permittivity given by:

$$\frac{\Delta \varepsilon}{\varepsilon} \propto 2 \frac{\Delta t_2}{t_2}$$

is substantially irrelevant.

**Ref. comment (page 6, line 18): For some reason you miss to report the state of the art on this combination made by several colleagues in the paragraph (see General Remarks).**

**Ref. comment (page 7, line 4): Again, you miss to relate this in a broader context. What about upward-looking radar systems in combination with GPS?**

**Ref. comment (page 19, line 1): If you plan to resubmit, please discuss this in a broader context by using a broader spectrum of existing literature. Especially Schmid et al. (2014) will be very helpful for you.**

**Reply**: We are well aware about the very interesting application of upward looking GPR published in several papers. However, we did not mention such papers in our manuscript because the field conditions and the rationale of the measurement are totally different and hardly comparable; consequently we compared our results with those reported in similar experiments. In particular, upward looking GPR has been employed at fixed locations monitoring the time evolution of the snow layer (surface). In contrast, in our case we have a strong spatial variability in the properties of the basal reflector. Therefore the two methodologies are affected by different phenomena and error sources.

**References:**

Lee, J. E., Lee, G. W., Earle, M., & Nitu, R. (2015). Uncertainty analysis for evaluating the accuracy of snow depth measurements. Hydrology and Earth System Sciences Discussions, 12, 4157-4190.

Denoth, A., 1994. An electronic device for long-term snow wetness recording. Ann. Glaciol., 19: 104-106.

Robinson D.A., S.B. Jones, J.M.Wraith, D. Or and S.P. Friedman (2003), A review of advances in dielectric and electrical conductivity measurements in soils using Time Domain Reflectometry, Vadose Zone J., 2, 444-475.

Robinson, D. A., M. Schaap, D. Or and S.B. Jones (2005), On the effective measurements frequency of time domain reflectometry in dispersive and conductive dielectric materials, Water Resour. Res., 41, W02007, doi: 10.1029/2004WR003816.

Topp, C.G., M. Yanuka, W.D. Zebchuk and S. Zegelin (1988), Determination of electrical conductivity using Time Domain Reflectometry: soil and water experiments in coaxial lines, Water Resour. Res., 24, 945-952.

Von Hippel (1954), Dielectric and Waves, New York, John Wiley & Sons Inc., London, Chapman & Hall Limited.

Yanuka M., C.G. Topp, S. Zegelin and W.D. Zebchuk (1988), Multiple reflection and attenuation of Time Domain Reflectometry pulses: theoretical considerations for applications to soil and water, Water Resour. Res., 24, 939-944.

Mattei, E., Di Matteo, A., De Santis, A., Pettinelli, E., & Vannaroni, G. (2006). Role of dispersive effects in determining probe and electromagnetic parameters by time domain reflectometry. Water Resour. Res., 42(8).

---

## Referee Comment (RC2) · Anonymous Referee #2 · 5 Apr 2017

The manuscript "Uncertainty budget in snow thickness and snow water equivalent using GPR and TDR techniques" by Di Paolo et al. presents an approach to quantify uncertainties from radar determinations in snow depth and SWE. The authors compare measured two-way travel times (TWT) with density and snow depth measured conventionally and TDR point observations in snow pits. From inclusion of device specific errors, they assess GPR specific uncertainties. Such an approach is not novel but interesting. Several studies before assessed uncertainties in conversion of TWT to derive SWE and snow depth (Lundberg with several studies and Sundstroem et al. 2012 for instance). Due to several severe misinterpretations and mistakes conducted mainly during fieldwork, I consider this manuscript (MS) not being sufficient for publication in

The Cryosphere. I recommend to reject it. In detail, the following remarks prevent it from publication:

- As major misinterpretation, I consider the comparison of TDR data with bulk densities and bulk wave speeds. You mention several times in the MS that TDR is "the best estimator" for wave velocity (e.g. p14, L4ff). I consider this as being fundamentally wrong. In the presented work you never refer on snowpack stratigraphy. While the presented density measurements and the GPR data are integrated over the whole snowpack, TDR measurements are only valid for a certain point within the pit wall. Techel and Pielmeier (2012) describe the support of the Finnish Snow Fork (a similar measurement device, you used in the field) being at V = 47 cm^3 , while the area includes 7.5 x 2 cm^2. In consequence, the vertical extent of such a measurement is only +-1.6 cm. So here you compare permittivity determined at a specific depth with bulk conditions for either 30 cm or the whole snowpack (and you assume for homogeneous conditions for 30 cm below your point measurement). Such point measurements are most likely highly influenced by local stratigraphy such as crusts, density differences or as mentioned in the MS the presence of liquid water. However, considering the data presented in Fig. 4 (neglecting the wrong scale in the submitted version) clearly describes discrepancies between TDR and density data. For the case, that liquid water is actually present in the snowpack, your whole conversion scheme has to be revised. It is no longer valid to present an empirical wave speed model not accounting for the volume fraction of water. Summarizing, I disagree with the statement that TDR is your best estimator in three points:

1. you create compaction of the analyzed snow layer while sliding in the instrument (see Techel and Pielmeier (2011) and Kinar and Pomoroy (2015) describing such circumstances) 2. as described above you only measure point conditions vertically with a very limited support and a 30 cm vertical resolution is not adequate to account for bulk conditions (only 10 measurements for a >3 m snowpack; Figure 4). 3. As described by Heilig et al. (2015), TDR measurements to determine permittivity are scattering a lot

and do not represent the assumed accuracy summarized by e.g. Kinar and Pomoroy (2015)

- According to my understanding, you address the wrong uncertainties in your analysis. For the snow density, the uncertainties are not related to the dimensions of the cylinder (which you quantify to +-0.3 mm p7 L12ff). Here you should implement uncertainties in the filling of the cylinder, which is usually the largest component of uncertainty, together with the calibration of the spring scale with which you weigh the sample. Such spring scales usually have an uncertainty related with weight and air temperature. As a third point, I consider the parallax error as being larger than the uncertainty of 0.3 mm in total diameter. None of these points are addressed in your error analysis for density measurements. It is very common in fieldwork that you apply at least two measurements right next to each other to account for such uncertainties.

- I agree with referee #1 that a snow probe with 0.5 m resolution is absolutely inadequate for the analysis you derive. You mention an uncertainty just from depth readings of +-0.14 m. This is somehow very questionable in relation with the presented uncertainties of +-3% in SWE (Table 3). I realized that the uncertainty in snow depth is just one contributing factor for the resulting uncertainties in SWE for GPR data conversion, however, with a snow probe of 0.5 m resolution you just don't present any useful data set for comparison.

- Not a single figure, data set, measure of the prevailing meteorological conditions during data acquisition is presented. Didn't you record snow temperatures, air temperature, moisture or wind during your field acquisitions? Such data would support the current presentation tremendously. Either in terms of understanding how much liquid water may have been present in the snowpack by estimating the energy balance and correspondingly the prevailing melt or how much the pit walls and snow parameters altered during acquisition (no data on sheltered or exposed pit wall either, no data on temporal differences between pit measurements and radar transects).

- Another major point, I criticize, concerns fieldwork as well. You never quantify how "close" radar transects match pit walls. The term "passing close" is not an adequate quantification. Spatial variability in snow depth, density and liquid water content can become very large and might be a much bigger component contributing to uncertainty than indicated in this manuscript.

Other major points that should be addressed in a revised resubmission: - Quite often you reference not the correct publications for respective parameters or methods, e.g. the relative dielectric permittivity of ice given as 3.29 published by Pettinelli et al. (2003) is certainly at the far end of publications. Why didn't you cite Bohleber et al. (2012), who measured such parameters most accurately or any previous well established literature (Hobbs). There are many other publications out introducing GPR for snow depth measurements (e.g. Vickers and Rose (1973) for the first publication describing it and Harper and Bradford (2003) as a more relevant and earlier citation than the ones listed in your MS). It is not a good habit to prefer self citations.

- Koch et al. (2015) conducted a much more in detail comparison of various conversion schemes than just the two you present.

- The initial submission was not carefully revised and checked for errors, typos etc. Apart from the major sloppiness in Figure 4, you do not introduce RMS as abbreviation and several statements are rather unsupported (P2 L17, Kinar and Pomoroy (2015) describe in detail several tubes to measure both parameters at the same time, P3 L13ff, Sundstroem et al. (2012) and Lundberg with several papers presented uncertainty estimates for such issues). Figure 6 has commas as decimal separators.

- You should include a plot or a further description on the field data.

- While you review GPR and field measurements quite excessively (appears to me as a repetition of Kinar and Pomory (2015) for some paragraphs), me as a reader coming from a different perspective, I got no impression about the factor k and what it is referring to and how you can adjust between k=1 or k=2.

[Figure]

Citations, which are not listed in your references but mentioned in my review: Bohleber, Pascal, Norman Wagner, and Olaf Eisen. "Permittivity of ice at radio frequencies: Part I. Coaxial transmission line cell." Cold Regions Science and Technology 82 (2012): 56-67.

Heilig, A., C. Mitterer, L. Schmid, N. Wever, J. Schweizer, H.-P. Marshall, and O. Eisen (2015), Seasonal and diurnal cycles of liquid water in snow—Measurements and modeling, J. Geophys. Res. Earth Surf., 120, 2139–2154, doi:10.1002/2015JF003593.

Hobbs, P. V. "Ice physics, 837 pp." (1974).

Koch, F., Prasch, M., Schmid, L., Schweizer, J., & Mauser, W. (2014). Measuring snow liquid water content with low-cost GPS receivers. Sensors, 14(11), 20975-20999.

---

## Editor Comment (EC1) · O Eisen (Editor) · 13 Apr 2017

Dear Federico Di Paolo and co-authors,

we now received two reviews for your manuscript. Both reviewers argue along the same line: your results are in general interesting but the present state of the submitted manuscript falls short of providing a sufficiently accurate description of all aspects of the measurements. These concern the measurements themselves, partly your approach, in particular the vertical resolution of snow probing, but also the interpretation and discussion in the context. Both reviewers point out significant deficiencies in the planning of the measurements and in data interpretation. Assuming in the most favorable case that you could write a thoroughly improved version, modifications would

be so extensive that they would require re-review and it would be preferable for you to present that as a new submission. However, your measurements cannot be improved, and these are also criticized so that I in fact seriously doubt that your data and their interpretation can be brought to an adequate level of rigor. For these reasons, I am sorry that I cannot encourage you to submit a revised version or resubmit your manuscript. My opinion is that additional careful and detailed measurements are required before your interesting ideas can be considered for publication in "The Cryosphere" again.

Regards, Olaf Eisen